# Winners' strategies: Comprehensive analysis and optimization of 2-point shots in 3x3 basketball using multi-criteria decision support analysis, on the example of two Olympic National Teams

Michał Nowak[1]*, Michał Skalik[1], Jakub Więckowski[2], Radosław Ciejpa[3], Artur Stolarczyk[4], Łukasz Oleksy[5]

1 Department of Physical Culture Sciences, Collegium Medicum named Dr. Władyslaw Biegański, Jan Dlugosz University in Czestochowa, Czestochowa, Poland, 2 National Institute of Telecommunications, Warsaw, Mazowieckie, Poland, 3 Basketball Sports Club "Dwójka" Kamienica Polski, Kamienica Polska, Poland, 4 Department of Orthopaedics and Rehabilitation, Medical Faculty, Medical University of Warsaw, Warsaw, Poland, 5 Department of Orthopedics, Traumatology and Hand Surgery, Faculty of Medicine, Wroclaw Medical University, Wroclaw, Poland

* edukacjawsporcie@icloud.com

## Abstract

The study presents an innovative approach to analyzing and optimizing 2-point shooting strategies in 3x3 basketball, using newly defined parameters and multi-criteria decision analysis (MCDA). A comparative analysis of the results of the Polish and Serbian national teams from the Olympic Games in Tokyo 2020 allowed us to identify key factors influencing the effectiveness of shots, such as the location of the shot, interaction with the defender, and the time of the action. It was shown that the Serbian team was more effective, which translated into a better result. Differences in shooting strategies were identified, especially in the crucial second half of the match, emphasizing the importance of tactical flexibility. Multi-criteria modeling showed that two factors – the effectiveness of shots and the average duration of an action – could be used to optimize game strategy. To leverage these findings, we emphasize the need for individualized training measures focused on generating shooting opportunities for oneself and teammates, which in turn increases decision-making effectiveness in key moments (e.g., when trailing in score). 3x3 basketball offers a unique opportunity to analyze strategies that rely more on cooperation, creativity, and improvisation than on the strictly defined tactical systems of 5x5. Research into this format can contribute to a deeper understanding of decision-making processes, shooting efficiency, and adaptive responses of players in high-intensity situations, which may also apply to other fast-paced team sports. This study proposes to continue the research with larger numbers of teams to develop new models for a more detailed understanding of shooting efficiency in this discipline, including the influence of fatigue, tactical adjustments, and partial automation of measurements. These further

**Data availability statement:** The input data sets and the code used for the analysis have been published in the open repository on Zenodo at DOI https://doi.org/10.5281/zeno-do.14021214. The online support materials are divided into three parts: A) MCDA analysis, B) raw input data sets (Excel) and parameter description (Word), and C) sample raw video.

**Funding:** The author(s) received no specific funding for this work.

**Competing interests:** The authors have declared that no competing interests exist.

studies will fill existing research gaps and positively impact the development of this new Olympic discipline. This is one of the first studies to apply multi-criteria decision analysis to 3x3 basketball tactics.

## Introduction

3x3 basketball is a relatively new sport that originated from the street game Street-ball, popular in the United States since the 1920s. The transformation of this form of entertainment into an organized sport and the organization of international tournaments, such as the Adidas Streetball Challenge in 1992, contributed to its popularization. The discipline quickly gained worldwide recognition; in 2018–2019, the number of tournaments increased by over 30% [1]. The International Olympic Committee (IOC) included 3x3 basketball in the Olympic program, and its debut took place at the Tokyo Games in 2020 (played in 2021 due to the pandemic) [2]. 5x5 basketball is very well-researched in physiology, psychology, technique, tactics, and match statistics. Numerous scientific publications confirm this [3–10]. 3x3 basketball, with its unique dynamics and different scoring system, creates new opportunities in research and training while shaping players' shooting preferences [11].

The game is played on a smaller court, and with a shortened action time (12 seconds instead of 24), the competition becomes faster and more intense. The increased frequency of interaction with the opponent emphasizes the importance of individual technique, such as dribbling or playing 1-on-1, because these skills are more difficult to use in a more crowded environment. This requires players to have great physical strength, exceptional physical condition, quick decision-making, and tactical flexibility. The lack of clear positions and the constant rotation of roles between attack and defence means that each player must be versatile - both as a shooter, playmaker, and defender. This increases the level of unpredictability of the game and requires greater concentration [12]. 3x3 basketball is characterized by short periods of effort interspersed with moments of rest, e.g., during player substitutions or free throws [13]. It requires intense anaerobic effort and more frequent contact with the ball [14–16]. Specific movements, such as jumping, acceleration, and stopping over short distances, elicit strong physiological responses, and frequent changes of direction are more intense than in 5x5 basketball [17–19]. The game lasts from 15 to 20 minutes, and the presence of only one substitute player means that players spend an average of 75% of the game time on the court [12]. The average heart rate is 152 beats per minute, and the maximum heart rate is 186 beats per minute [18].

In many ways, however, 3x3 basketball resembles 5x5. Studies on a group of 15 players aged 16–17 showed that parameters such as heart rate, oxygen saturation, and hemoglobin level did not differ significantly between these forms of play [20]. In 3x3 tournaments, two or three matches are often played per day, which requires rapid muscle regeneration. Studies have shown that after a single match, muscle contractility decreases, but players recover quickly and are ready for the next games [21,22]. In addition to physiology and motor skills, the analysis also included game efficiency – tactics, game patterns, ball possession efficiency, and selectivity

of 1- and 2-point shots. The results showed that winning teams take more free throws and use screens and pick-and-roll actions more often [23,24]. In 3x3, the number of long-distance shots (from outside the 6.75 m line) is higher, but their efficiency is lower [25,26]. Offensive efficiency is around 27.5% in 3x3, which is lower than in 5x5 basketball (35%).

From a scientific perspective, 3x3 basketball offers a unique opportunity to analyze strategies that rely more on cooperation, creativity, and improvisation skills than on strictly defined tactical systems characteristic of 5x5. Research on this format can contribute to a deeper understanding of decision-making processes, shooting efficiency, and adaptive reactions of players in high-intensity situations, which can also be applied to other fast-paced team sports. At the same time, the literature lacks detailed studies on the tactical behaviour of players in 3x3 basketball. Previous work has focused mainly on the physical and physiological aspects of this discipline; our work, on the other hand, focuses on the analysis of throwing tactics, which is a significant research gap.

### Aim of research

The presented state of research on the problems of the 3x3 game is not exhaustive. There is a need to continue to discover the key elements that influence winning or losing. Taking into account that one of the most contrasting differences in the rules of 3x3 and 5x5 games is the point counting system (2-point shot vs. 3-point shot), the authors of this work decided to cover a broader analysis of two-point throws by Serbian (SRB) teams (7 wins - 0 losses) vs. Poland (POL) (2–5), during the group stage of the tournament held as part of the Olympic Games in Tokyo 2020.

Research questions:

**1. To what extent does the effectiveness of 2-point shots determine winning the game?**

**2. What parameters have the most significant impact on the effectiveness of 2-point shots?**

Detailed questions were also asked:

• From which areas of the pitch are most 2-point shots taken?

• Does the length of the shooting action affect the effectiveness of 2-point shots?

• What impact does the active participation of the defender in the shooting action (open-closed position) have on the effectiveness of 2-point shots?

• Is there a difference in effectiveness between 2-point shots after the dribble and after a pass from a partner?

• Does the current match result affect the effectiveness of 2-point shots?

• Does the effectiveness of 2-point shots change as the match progresses (1–5 minutes/ remaining time)?

• What was the effectiveness of the teams' play in each part of the game taking into account points scored after dribbling and after passing?

• Can the multi-criteria decision analysis (MCDA) model identify areas of the game that need improvement?

• Based on a multi-criteria analysis of the teams' performance ratings in individual games, can it be identified which aspects of the game were most important?

## Materials and methods

### Subjects and protocol

The study analyzed the 2-point shooting situations of Polish and Serbian national team players during the 2020 Tokyo Olympic Games. The players were selected by national team coaches based on their highest qualifications, which

ensures that the group studied is the sports elite in their countries. The composition of the national teams, therefore, reflected the highest level of skill available in a given country at a given time. To facilitate reading the context of events in the 3x3 game, an analysis of video material collected during the tournament's group phase held as part of the Olympic Games in Tokyo in 2020 was used. Seven matches of the Serbian national team (7 wins - 0 losses) and seven games of the Polish national team (2 wins -5 losses) were analyzed. These games were available on a streaming platform for a subscription fee. The Serbian team took 3rd place in the final classification of the tournament; the Polish squad did not advance past the group stage. Video analysis was performed using ProTrainUp LiveTagPro (Version 1.8.3), which enables accurate tagging and tracking of on-court events. Additionally, each clip was analyzed by two league-licensed coaches with 15 years of basketball experience, ensuring high accuracy and reliability of results. Each coach worked separately, and in case of discrepancies, they decided together. Video analysis allows for detailed tracking of the action in high-intensity matches, which is crucial in sports such as 3x3 basketball. However, this method has limitations, such as dependence on the recording quality and limited access to data on the players' physical loads. In future studies, it is worth considering complementing this method with motion measurement technologies, such as local positioning systems.

## Definition and identification of parameters

For the study, a new division of the playing field was created. The researchers used the following parameters: Throw zone (wings, half-wings, center) - Section 1- Section 5 (S1- S5); Distance from the line 6.75 m (0–50 cm; 51–100 cm; 101–150 cm and more than 151 cm) at the moment of making the throw Line 0 - Line 3 (L0 - L3); Time of making the throw in a given action in time intervals (1–4 sec.; 5–8 sec.; 9–12 sec.); Type of throw (after Dribbling "D" or after a Pass from partner "P"), Open position "O" and Closed position "C". Successful throws are marked in green, missed throws are marked in red. "O" - the open position means that the defender's behaviour did not affect the throw (he was more than 1.5 m away from the throwing player). "C" - closed position means that the defender absorbs the thrower by being close, below 1.5 m, and interfering with the throw with his hands. "P" - obtaining a shooting position thanks to a good pass (assist) from a partner. "D" - is a shooting position obtained after 1-on-1 play, pick and roll, or other screens. The analysis took into account the relationship between the throw during the game and the match score in the following categories: unfavorable result - negative (-), draw - zero (0), and point advantage over the opponent - plus (+). Each analysis was performed in the first 5 minutes of the game and in the time until the end of the match. Fig 1 (A and B) shows sample parameters in all possible versions.

## Ethical statements

The paper's authors declare that the analysis was based on video material. It was publicly available via television broadcast to the general public. The analysis did not include personal data or private information. The study did not involve humans or animals. The study did not affect individuals and was completely non-invasive. Based on these premises, it was concluded that ethical consent was not necessary to conduct the analysis.

## Statistical analysis

### Basic statistics

Microsoft® Excel for Mac, version 16.81 (24011420), available as part of the Microsoft 365 subscription, was used for statistical analysis. Basic statistical functions available in the Excel package were used to perform the analysis, which allowed for calculating sums and percentages for the set data. This approach enables basic quantification and presentation of data that can be used for further conclusions and interpretation of results. Using Excel as a widely available tool and recognized as a standard in statistical analysis provides access to the necessary tools to perform basic statistical

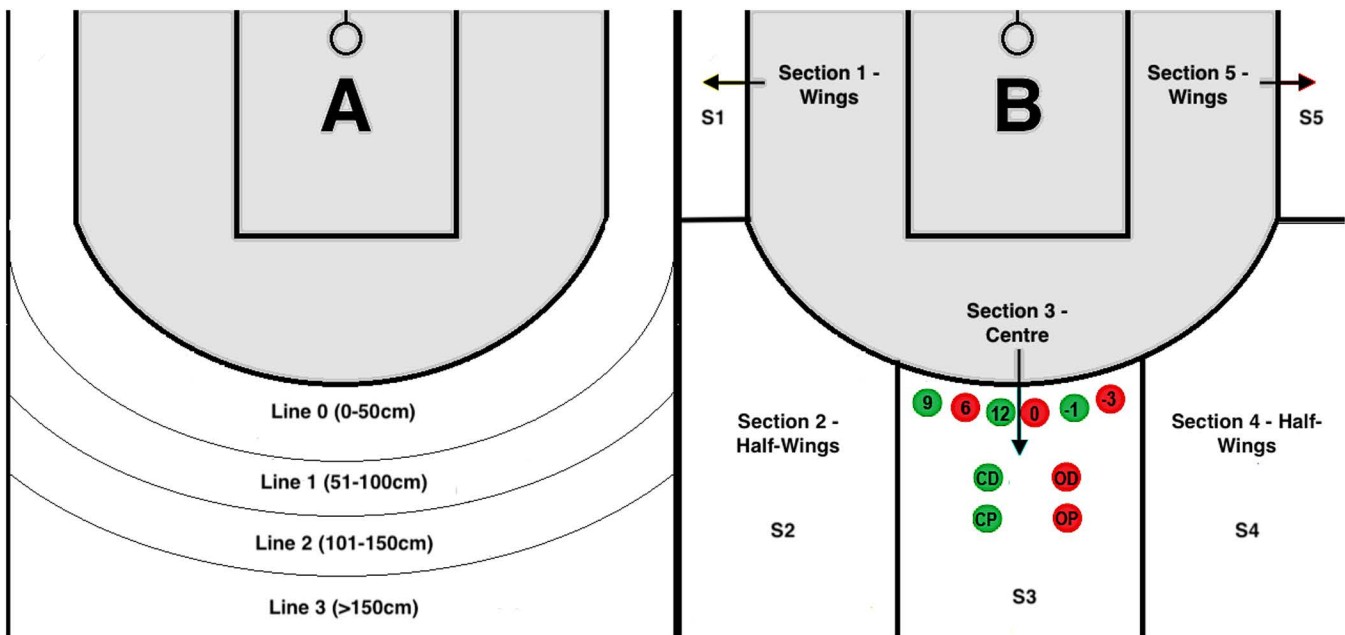

**Fig 1. Visualization of analyzed parameters.** Figure "A" (left side) shows the division of the playing field space about 2-point shots related to the distance from the 6.75 m line. Line 0 (0-50 cm), Line 1 (51-100 cm), Line 2 (101 -150 cm), Line 3 (over 150 cm). Figure "B" (right side) shows the division of the playing field space about the place for taking 2-point shots. Section S1 and S5, "Wings"; Section S2 S4 "Half Wing"; and Section S3 "Centre". In the Centre of Section S3, representative symbols (circles) corresponding to the analysis of the research context are placed: 9, 6, 12 (seconds of action); 0,-1,-3 (difference in result); CD (Closed after Dribbling); CP (Closed after Pass); OD (Open after Dribbling); OP (Open after Pass). Filling the circles with Red means the throw was unsuccessful, and Green means it was successful. The zone marked in grey was not included in the analysis.

calculations, such as sum, average, median or percentage calculation. This makes the results easy to reproduce and verify by other Excel users, which is essential for the transparency and replicability of the study.

## Multi-criteria decision analysis

A multi-criteria decision analysis approach was used to analyze the tested group of athletes in terms of results, which allows for the assessment of alternative decisions based on many criteria, often conflicting goals. MCDA methods are tools that allow you to evaluate the tested alternatives in a structured way and indicate their ranking based on the obtained preferences. They are used in sports-related issues, including in the assessment of athletes, and described in more detail in the work [27]. Statistics from the match were used in further evaluation of teams' performance regarding two critical parameters: the accuracy of throws ($C_1$) and mean action time ($C_2$). These two criteria were used to define the Multi-Criteria Decision Analysis (MCDA) decision model, which was used to assess the teams. The model was built upon the Characteristic Objects Method (COMET), which formal notation is presented in Sałabun et al. (2018) [28]. The structure of the decision model used in the assessment process is presented in Fig 2. As the result of the assessment, the preference scores ($P$) are obtained, producing values from the range 0 and 1, where greater values indicate better performance.

The data gathered from the match was used to determine the initial decision matrix used for evaluation purposes. The teams' performance was grouped by the two halves of the game and by the type of action from which the throw was made (passing or dribbling). It allowed us to identify eight alternatives, four for each country. Then, based on the statistics, the accuracy ($C_1$) and mean action time ($C_2$) were indicated for each team variant. The determined decision matrix is presented in Table 1.

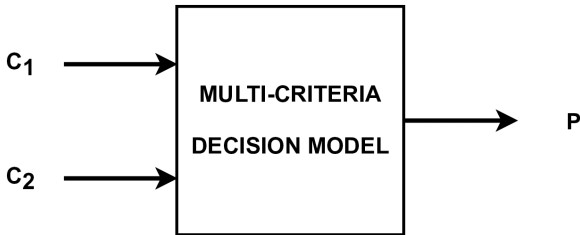

**Fig 2. The structure of the decision model is defined for the evaluation performance of the analyzed teams.** C1 - accuracy of throws; C2 - action time; P- preference scores.

**Table 1. A decision matrix is used to evaluate the performance of teams regarding different parts of the game and different ways of making a throw.**

| Ai | Team | Accuracy (C1) | Mean Action Time (C2) |
|---|---|---|---|
| $A_1$ | Poland (passing 1–5 minutes) | 0.290323 | 5.258065 |
| $A_2$ | Poland (dribbling 1–5 minutes) | 0.350000 | 7.050000 |
| $A_3$ | Poland (passing 6–10 minutes) | 0.320000 | 5.280000 |
| $A_4$ | Poland (dribbling 6–10 minutes) | 0.269231 | 6.846154 |
| $A_5$ | Serbia (passing 1–5 minutes) | 0.368421 | 5.052632 |
| $A_6$ | Serbia (dribbling 1–5 minutes) | 0.125000 | 8.750000 |
| $A_7$ | Serbia (passing 6–10 minutes) | 0.416667 | 5.250000 |
| $A_8$ | Serbia (dribbling 6–10 minutes) | 0.500000 | 7.357143 |

Ai - (A1- A8) - Combination of throws after passing and after dribbling separately for Serbian and Polish national teams divided into time intervals (1–5 minutes) and (6–10 minutes).

## Results

### Efficiency of shooting strategies and match outcomes

A comparative analysis of all shooting situations (2 points) of the Polish and Serbian teams in the group phase of the Olympic Games, divided into periods from 1 to 5 minutes and from 6 to 10 minutes, taking into account sections on the pitch and the distance from the 6.75 m line, was presented on Figs 3–5.

Fig 3 shows the context of projections in the set of parameters "O" or "C" and their combinations "P" and "D". POL made the same number of shots in both observed periods, i.e. (51/51) and achieved similar effectiveness (31%/29%). SRB represented a different strategy by making fewer shots (35/26). The first part shows slightly lower, and in the second, much higher effectiveness (26%/46%).. The average attempts per game were 14.57 for Poland and 8.71 for Serbia. The effectiveness of both teams was similar - 30.39% for Poland and 34.43% for Serbia.

• **Sections (surface on the pitch 1-5).** Section 2 (left side) and section 3 (middle), regardless of the team and game time, were the places most frequently chosen by players to take shots. In the first part, the total was 38 for POL/SRB 28 throws. In the second half, POL 35/SRB 19 shots. The effectiveness remained similar: POL (39%/34%) and SRB (21%/42%). The most uncomfortable place on the pitch turned out to be section 5. POL made 4 throws from this place, all of them missed, and SRB made 2 throws, and 1 was successful. In both teams, throws from the sections closest to the basket (S1 and S5) accounted for the smallest percentage of all throws - 11.76% for Poland and 8.20% for Serbia. The middle section (S3) was used most often for shots by both teams, with a success rate of 32.56% for Poland and 38.46% for Serbia, which was not their highest efficiency result. When the central section (S3) was omitted, both teams preferred

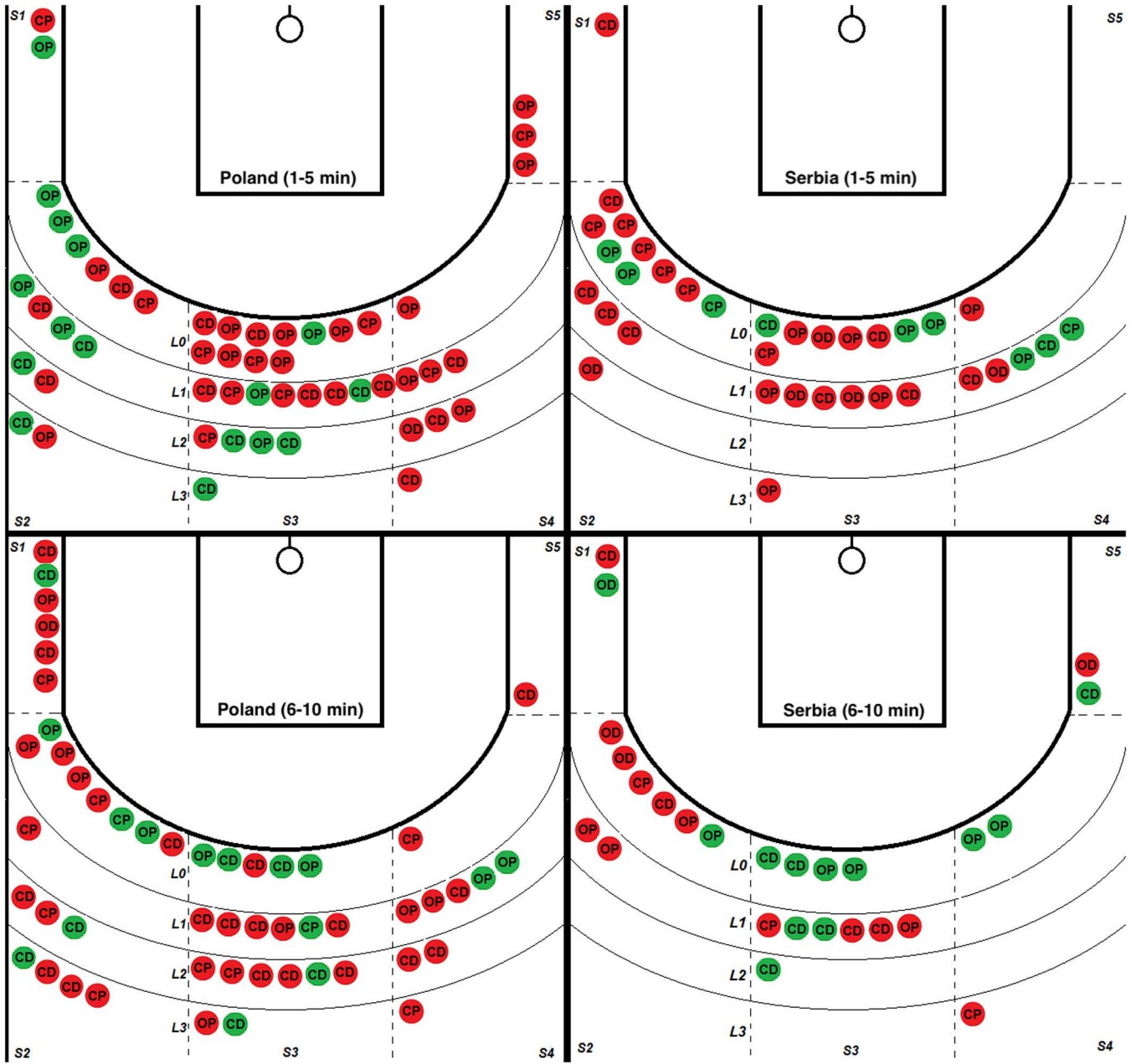

**Fig 3. Comparison of all 2-point shots made in the group phase of the Olympic tournament by the Polish and Serbian teams.** Successful - green, Unsuccessful red; two periods (1-5 min) and (6-10 min) with sections marked on the pitch (S1-S5) and distance from the line (LO-L3) 6.75 m. Closed after Dribble (CD); Closed after Pass (CP); Open after Dribbling (OD); Open after Pass (OP).

shots from the left side of the field, mainly from zone 2, where Poland took 30 shots and Serbia had 21 shots. The effectiveness of these throws was 43.33% for Poland and 19.05% for Serbia, respectively, being the best result for Poland and the worst for Serbia compared to other sections (1–5).

• **Distance from 6,75 line.** The boundary distance for the SRB representation was the space bounded by the L1 line. The percentage of all throws in the first and second parts was equal (94%/92%). For comparison, POL (74%/65%) for

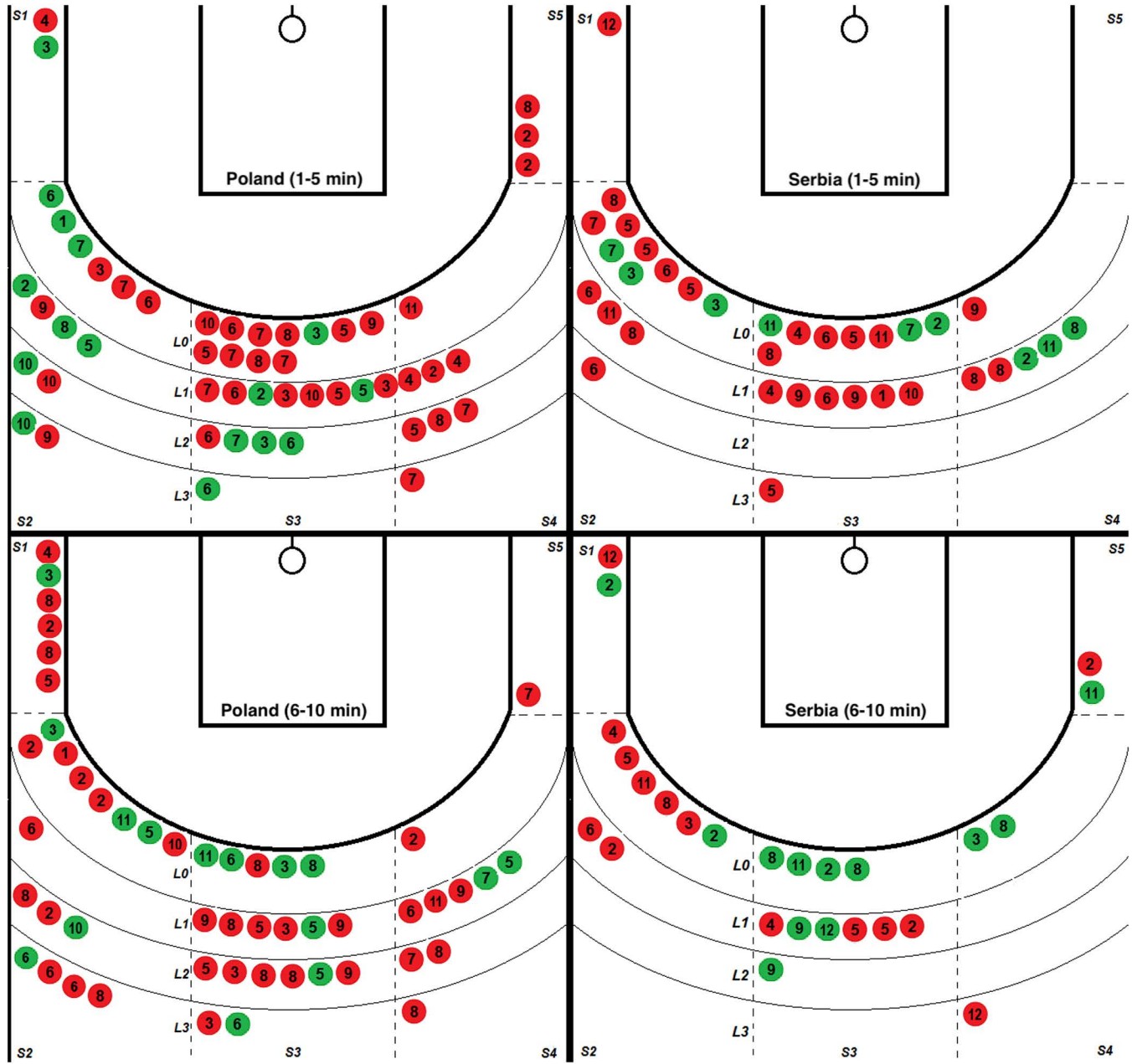

**Fig 4. Comparison of all 2-point shots made in the group phase of the Olympic tournament by the Polish and Serbian national teams.** Successful - green, Unsuccessful red; two periods (1-5 min) and (6-10 min) with sections marked on the pitch (S1-S5) and distance from the line (LO-L3) 6.75 m. The numbers (1-12 sec) show the second in which the shot was made.

the same distance. Comparing the effectiveness of POL on the example of the distance L0 - L1 relative to L2 - L3, it is (26%/33%) to (46%/22%). A significant difference between the teams was the number of shots taken outside line 2 (L2), where Poland made 31 shots (30.39% of all Polish shots) and Serbia only 4 (6.56% of all Serbian shots). For Poland, the throw distance (L0 - L3) did not significantly impact the effectiveness, which oscillated between 29.55% and 36.36%. Serbia most often threw from the closest line (L0), which accounted for 57.37% of all its throws, with a high effectiveness of 42.86%. Table 2 presents a summary of the data.

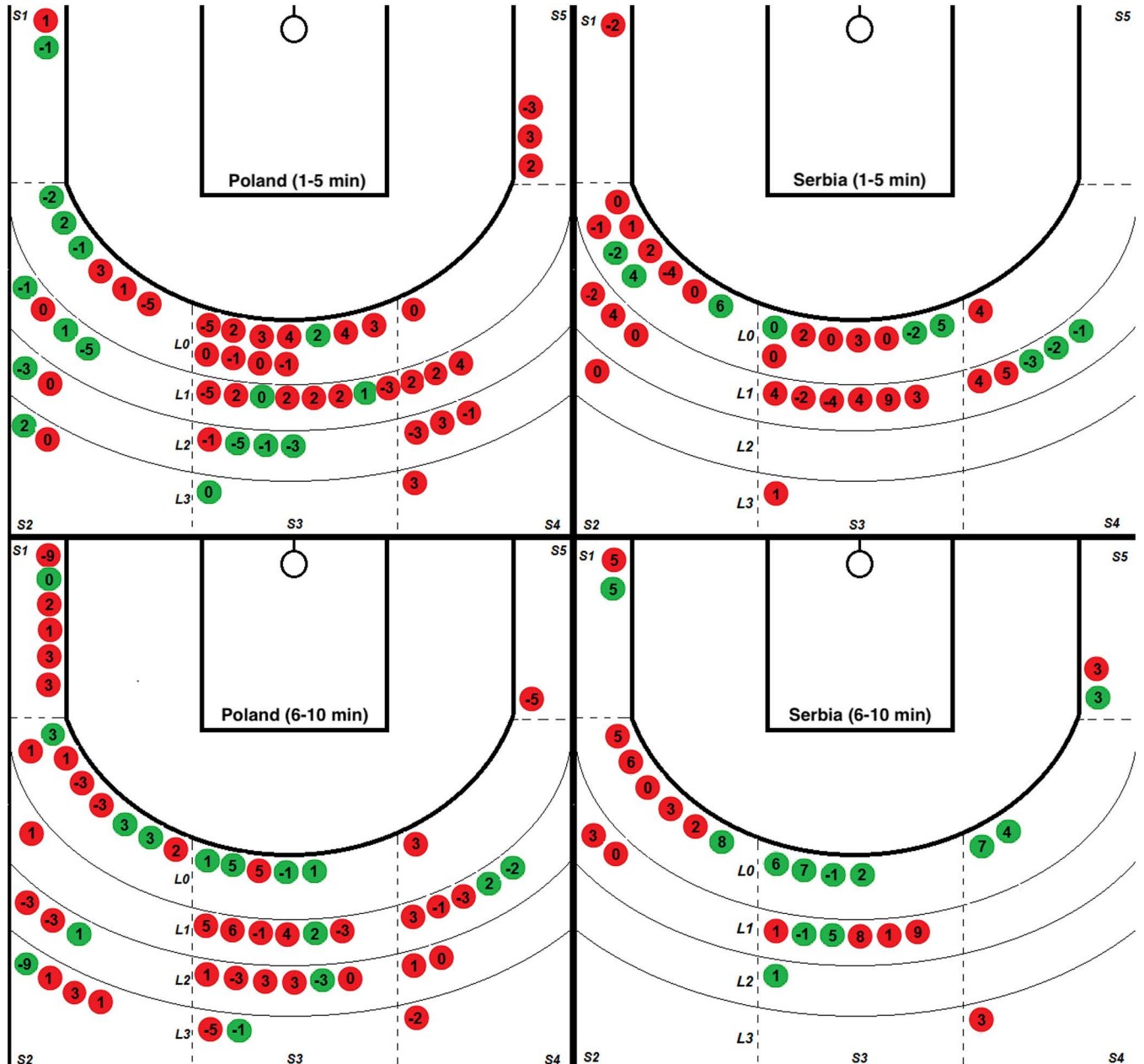

**Fig 5. Comparison of all 2-point shots made in the group phase of the Olympic tournament by the Polish and Serbian teams** Successful - green, Unsuccessful – red; two parts (1-5 min) and (6-10 min) with the marks marked in pitch in sections (S1-S5) and the distance from the line (LO-L3) 6.75 m. The numbers in the circles show the size of the difference in the result when taking the shot. A negative number is "the team was losing", a positive number is "the team was winning", and a zero is "the result was a draw.".

## Parameters (OP, PD and CP, CD)

The highest accuracy of throws was obtained from the OP position and was respectively for POL (50%/56%) and SRB (43%/38%) in the first and second part of the analyzed game time. POL performs better than SRB regarding prepared shots from the OP position, i.e., in the first and second parts, there were (21/14) actions compared to SRB (11/9). The OD position was too difficult. Effectively creating space for throwing was impossible - only once did an SRB player

**Table 2. Summary of 2-point shots by the Polish and Serbian teams in the Tokyo 2020 Olympic Games tournament in total.**

| The name of the parameter | Symbol | POLAND | | SERBIA | |
|---|---|---|---|---|---|
| | | Shots made/throws on target (number) | Shots made/throws on target (%) | Shots made/throws on target (number) | Shots made/throws on target (%) |
| Section 1 | S1 | 2/8 | 25,00% | 1/3 | 33,33% |
| Section 2 | S2 | 13/30 | 43,33% | 4/21 | 19,05% |
| Section 3 | S3 | 14/43 | 32,56% | 10/26 | 38,46% |
| Section 4 | S4 | 2/17 | 11,76% | 5/9 | 55,56% |
| Section 5 | S5 | 0/4 | 0,00% | 1/2 | 50% |
| Sum (on target/missed) | | 31/102 | 30,39% | 21/61 | 34,43% |
| Line 0 | L0 | 13/44 | 29,55% | 15/35 | 42,86% |
| Line 1 | L1 | 8/27 | 29,63% | 2/22 | 22,73% |
| Line 2 | L2 | 6/20 | 30,00% | 1/2 | 50,00% |
| Line 3 | L3 | 4/11 | 36,36% | 0/2 | 0,00% |
| Sum (on target/missed) | | 31/102 | 30,39% | 21/61 | 34,43% |

S1 - S5 - place of the throw (divided into sections) and the distance from the 6.75 m line (divided into lines) L0 - L3. Green - the most shots made or the highest percentage of success. Red - the lowest number of shots made or the lowest percentage of success.

successfully do it out of 11 attempts in both teams. Throwing from the CP position turned out to be an equally difficult task. Both teams had 2 shots to the basket in the first and second halves. POL (2/0) and SRG (0/2) with effectiveness at the level of POL (25%/0%) and SRB (0%/18%). Players equally often decided to throw from the CD position in a similar way as from the OP position. In the first and second parts, POL made (19/25) shots with a percentage of hits (37%/29%) in sections 2 and 3, and SRB (11/10) shots with a rate of hits (18%/60%) most of them in section 3. Analysis of the percentage of different types of shots shows that both Poland and Serbia had similar preferences: shots after dribbling from a closed position (CD) (41/21), open positions after passing (OP) (35/20), closed positions after passing (CP) (21/11) and open positions after dribbling (OD) (2/9).

• **Time periods in the attacking game.** Serbia achieved the highest efficiency in quick actions (1–4 seconds), regardless of position (open - 46.67%, closed - 50%). Serbia also excelled in long actions (9–12 seconds) from closed positions, achieving an efficiency of 41.18% (7/17). Poland made the most shots in the average time (5–8 seconds) from both open and closed positions. However, the highest number of throws does not translate into the highest effectiveness. The biggest differences between Poland and Serbia in the effectiveness of throws in different time intervals are visible in throws from closed positions. Table 3 contains summary data on throws in individual time intervals, taking into account open and closed positions.

Fig 4 shows how often and how effectively teams made shots in different seconds of action (fast: 1–4 seconds; medium: 5–8 seconds; long: 9–12 seconds). At the beginning of the match (1–5 min), both teams mostly threw in the average time (5–8 seconds) - Poland (POL) on 8/27 and Serbia (SRB) on 3/19. Poland was almost twice as effective in this phase (POL 30% vs SRB 16%). The highest throwing efficiency in both teams occurred at different moments in the first (1–5 min) and second (6–10 min) part of the match. In the first phase, for POL in fast actions (1–4 sec.), it was 40% (6/15), and for SRB, 57% (4/7). In the second phase, in long actions (9–12 sec.), POL achieved 33% (3/9) and SRB 63% (5/8). When comparing the effectiveness of throws in different time intervals between the two phases of the game, for Poland, there is a decrease of 19% in quick actions, an increase of 2% in medium actions and an increase of 11% in long actions. On the other hand, Serbia recorded a decline of 17% in quick stocks, an increase of 22% in medium stocks and 41% in long stocks. In fast actions, Poland made 6.5 times more shots than Serbia, but the effectiveness of these shots was only 15.38%, which gives a difference in favor of Serbia of 34.62%. In the average time, Poland took 2.37 times more shots

**Table 3. Summary of 2-point shots of the Polish and Serbian teams in the group stage of the Tokyo 2020 Olympic Games tournament.**

| The name of the parameter | Symbol | POLAND | | SERBIA | |
|---|---|---|---|---|---|
| | | Shots made/throws on target (number) | Shots made/throws on target (%) | Shots made/throws on target (number) | Shots made/throws on target (%) |
| Open after Pass | OP | 15/35 | 42,86% | 10/20 | 50,00% |
| Open after Dribbling | OD | 0/2 | 0,00% | 1/9 | 11,11% |
| Closed after Pass | CP | 2/21 | 9,52% | 2/11 | 18,18% |
| Closed after Dribbling | CD | 14/44 | 31,82% | 8/21 | 38,10% |
| Sum (on target/missed) | | 31/102 | 34,43% | 21/61 | 34,43% |
| Open position in time from | O(1–4sec) | 7/16 | 43,75% | 7/15 | 46,67% |
| Open position in time from | O(5–8sec) | 7/17 | 41,18% | 4/11 | 36,36% |
| Open position in time from | O(9–12sec) | 1/4 | 25,00% | 0/3 | 0,00% |
| Closed position in time from | C(1–4sec) | 2/13 | 15,38% | 1/2 | 50,00% |
| Closed position in time from | C(5–8sec) | 10/38 | 26,32% | 2/16 | 12,50% |
| Closed position in time from | C(9–12sec) | 4/14 | 28,57% | 7/17 | 41,80% |
| Sum (on target/missed) | | 31/102 | 34,43% | 21/61 | 34,43% |

OP - open position after passing; OD - open position after dribbling; CP closed position after passing, CD - closed position after dribbling. The three time periods in the attacking (1–4 sec), (5–8 sec), and (9–12 sec) for the O - open and C - closed positions. Green - the most shots made or the highest percentage of success. Red - the lowest number of shots made or the lowest percentage of effectiveness.

than Serbia, with an efficiency advantage of 13.83% in favor of Poland. The number of throws was similar in long actions, but Serbia was more effective by 12.61%. It should be emphasized that Poland achieved the highest efficiency in all time intervals in shots from closed positions (28.57%).

• **Point advantage "+" – Draw "0" - Point deficit "-".** Fig 5 shows how successful and missed throws are distributed depending on the match result. In the first part of the match, both teams showed similar effectiveness in shots taken when the result was unfavorable for them (point deficit "-"), 47% for Poland, and 50% for Serbia, respectively. However, the situation changed in the second part, when the result decided about winning or losing. Poland fired a total of 18 shots with a 28% effectiveness rate, while Serbia fired only two shots, but both were on target, which gives a 100% effectiveness rate. Table 4 shows the numbers and effectiveness, comparing both teams when they played with an advantage, with a draw or when they were chasing the result. The SRB team demonstrated the lowest level of effectiveness in a draw, with only 10% effectiveness with ten throws. Even though both teams were the most effective during negative results, the SRB team was better than the POL by 16.01%. The result obtained was 53.85%.

## Multi-criteria decision analysis

The analyzed basketball teams were assessed using the determined decision model. Since the MCDA approach is characterized by the possibility of defining the importance of criteria in the problem, different sets of criteria weights were evaluated to examine the influence of various importance of decision factors on teams' performance scores. Five distinctive criteria weights sets were used, namely 1) 30% importance of accuracy and 70% importance of mean action time, and 2) 40% importance of accuracy and 60% importance of mean action time 3) equal importance of accuracy and mean action time, 4) 60% importance of accuracy and 40% importance of mean action time, and 5) 70% importance of accuracy and

**Table 4. Summary of 2-point shots of the Polish and Serbian teams in the group stage of the Tokyo 2020 Olympic Games tournament in total.**

| The name of the parameter | Sym-bol | POLAND | | SERBIA | |
|---|---|---|---|---|---|
| | | Shots made/throws on target (number) | Shots made/throws on target (%) | Shots made/throws on target (number) | Shots made/throws on target (%) |
| Point advantage | "+" | 14/54 | 25,93% | 13/38 | 34,21% |
| Draw | "O" | 3/11 | 27,27% | 1/10 | 10,00% |
| Point deficit | "-" | 14/37 | 37,84% | 7/13 | 53,85% |
| Sum (on target/missed) | | 31/102 | 30,39% | 21/61 | 34,43% |

The analysis included a throw made in the event of a "+" point advantage over the opponent, In the event of a draw "0", and an unfavorable result "-". Green means the most shots made or the highest percentage of success. Red is the lowest number of shots made or the lowest percentage of effectiveness.

30% importance of mean action time. From evaluations, it was possible to indicate the final preference scores regarding the relevance of different criteria and, based on that, define the ranking. The obtained results are presented in Table 5.

The range of preference results is presented as a percentage with lower and upper limits. (A1- A8) - Combination of throws after passing and after dribbling separately for Serbian and Polish national teams divided into time intervals (1–5 minutes) and (6–10 minutes) more in Table 1.

Table 6 analyzes the impact of hypothetical changes proposed to increase throwing efficiency on the entire team's performance. 5%, 10%, and 20% change in effectiveness from baseline. Data is presented for two critical aspects of the game: effectiveness in the second half of the match and effectiveness of shots after passing. For each version of the changes, differences in the model's preferences were examined, which allowed us to see how they could potentially - hypothetically - increase the chances of the team's final win. The results of the SRB A7 team showed the most significant improvement in model ratings at the highest level of improvement (20%), which confirms the team's dominance in the analyzed scenarios.

## Discussion

### The impact of the effectiveness of 2-point shots on the game result

The effectiveness of attempted 2-point shots is very important for the final result in 3x3 basketball games. This is confirmed by our study on the example of the SRB and POL teams during the 2020 Olympic Games in Tokyo. The SRB team was more effective in 2-point shooting and achieved a better final result. It should be emphasized that in this case, precision in throws may decide about victory. Better 2-point shooting accuracy increases the chances of winning, consistent with previous research (Conte et al., 2019; Erculj et al., 2020) [23,25]. This observation highlights that 2-point shooting training should be a priority for coaches and players in addition to game strategy (Tactic) and must also be an important point in the national team coach's work. Nowadays, players make ranged decisions a throw equally often in 3x3 games as in 5x5 games. The weight ratio of these throws in a 3x3 game (2:1) compared to a 5x5 game (3:2) may encourage such a tactical strategy [11]. However, it should be remembered that individual players' technical training level undoubtedly influences this [29].

### The importance of position and time of action for the effectiveness of throws

The research revealed that new definitions of pitch markers proposed by the authors (look at chapter research methodology), such as place on the pitch, length of the action, and interaction with the defender, are of key importance and big application potential. For example, the SRB team more often chose advantageous shooting positions and effectively exploited moments when the opponent's defence was weaker or the score pressure was lower, which increased their shooting efficiency. In contrast, the POL team had greater difficulty obtaining clean positions (OP), which negatively

**Table 5. Preference scores and ranking positions of teams including different distribution of criteria importance.**

| Importance | C1-C2 | | C1-C2 | | C1-C2 | | C1-C2 | | C1-C2 | |
|---|---|---|---|---|---|---|---|---|---|---|
| | 30% - 70% | | 40% - 60% | | 50% - 50% | | 60% - 40% | | 70% - 30% | |
| Ai | P | Rank | P | Rank | P | Rank | P | Rank | P | Rank |
| A1 | 0.818 | 4 | 0.804 | 4 | 0.692 | 4 | 0.678 | 5 | 0.567 | 5 |
| A2 | 0.495 | 6 | 0.520 | 6 | 0.529 | 6 | 0.555 | 6 | 0.565 | 6 |
| A3 | 0.833 | 3 | 0.834 | 3 | 0.729 | 3 | 0.729 | 4 | 0.624 | 4 |
| A4 | 0.482 | 7 | 0.453 | 7 | 0.449 | 7 | 0.421 | 7 | 0.417 | 7 |
| A5 | 0.912 | *1* | 0.912 | *1* | 0.824 | *2* | 0.824 | *2* | 0.736 | 3 |
| A6 | 0.000 | 8 | 0.000 | 8 | 0.000 | 8 | 0.000 | 8 | 0.000 | 8 |
| A7 | 0.904 | *2* | 0.911 | *2* | 0.862 | *1* | 0.869 | *1* | 0.819 | *2* |
| A8 | 0.532 | 5 | 0.657 | 5 | 0.688 | 5 | 0.813 | 3 | 0.844 | *1* |

Criteria weights between $C_1$(importance of accuracy) - $C_2$ (importance of mean action): 1) 30% $C_1$ and 70% $C_2$; 2) 40% $C_1$ and 60% $C_2$; 3) equal importance of accuracy and mean action time; 4) 60% $C_1$ and 40% $C_2$; 5) 70% $C_1$ and 30% $C_2$. (A1- A8) - Combination of throws after passing and after dribbling separately for Serbian and Polish national teams divided into time intervals (1–5 minutes) and (6–10 minutes) more in Table 1. The preference scores (P). Ranking - according to the weighting preferences of various criteria.

**Table 6. Result of the impact of hypothetical changes proposed to increase throwing efficiency on the entire team's performance.**

| Improvement | 5% | | 10% | | 20% | |
|---|---|---|---|---|---|---|
| Ai | Lower | Upper | Lower | Upper | Lower | Upper |
| A1 | 0.581 | 0.596 | 0.581 | 0.625 | 0.581 | 0.683 |
| A2 | 0.589 | 0.599 | 0.613 | 0.635 | 0.660 | 0.705 |
| A3 | 0.640 | 0.657 | 0.640 | 0.688 | 0.640 | 0.753 |
| A4 | 0.440 | 0.444 | 0.463 | 0.471 | 0.509 | 0.524 |
| A5 | 0.737 | 0.774 | 0.737 | 0.810 | 0.737 | 0.884 |
| A6 | 0.012 | 0.029 | 0.025 | 0.059 | 0.050 | 0.118 |
| A7 | 0.833 | 0.861 | 0.833 | 0.903 | 0.833 | 0.986 |
| A8 | 0.844 | 0.869 | 0.844 | 0.893 | 0.844 | 0.944 |

affected their shooting efficiency. This observation is consistent with findings in the literature [12]. Possible reasons include slower ball movement during passes or differences in the players' one-on-one skills. In addition to technical skills, a tactical understanding of the game becomes very important, and the ability to adapt quickly to various forms of the opponent's defence may be crucial for effectiveness play. By incorporating these findings into training, coaches can equip players with new tactical and technical solutions, potentially changing the status quo., which is necessary to consider when preparing players for 3x3 competition. Coaches should also consider individual differences in how quickly players adapt to new tasks and how well they can independently solve challenges under time pressure when selecting players.

### Throwing strategies depending on the situation in the match

Research has shown that in the first 5 minutes of the game, there are no significant differences between strategies regarding the use of 2-point shot weapons. At this stage, teams demonstrate great flexibility and adaptability. They often modify their approach. Depending on the situation on the pitch, they try to effectively exploit the opponent's temporary weaknesses and score points from the situational advantage gained thanks to their attacking activity. The biggest differences in the approach to 2-point shots were observed in the so-called decisive moments of the game (6–10 min). The SRB team adjusted their shooting decisions more effectively, limiting the number of shots relative to the opponent, which

had a significant impact on their victories. During the decisive phase (6–10 min), the POL team maintained both the volume and percentage of two-point attempts at similar levels as in the early phase. However, this stability did not translate into a better outcome. Despite adhering to their game plan and showing comparable physical performance, this strategy proved ineffective, contributing to Poland's overall defeat. This strategy turned out to be ineffective, which could have contributed to poor final results. This indicates the need to create training situations in which not only individual skills but also the development of strategic thinking will improve flexibility in making decisions on the pitch. In particular, this should apply to difficult situations for the team (fatigue, point deficit, short time, pressing in defence, playing on the verge of foul, unfavorable referee decisions, etc.). This is consistent with Li, X et al. 2024[30]

**Multi-criteria analysis in game optimization**

From the results presented in Table 5, it can be seen that, in general, the different variants of the Serbia team were evaluated better than the Poland team, especially in the second part of the game when throws after passing were analyzed ($A_7$). However, it can also be seen that the Serbia team, in the first half of the game had the worst performance regarding throws made after dribbling ($A_6$). On the other hand, the Poland team was ranked relatively high regarding the second-half performance in throws after passing, where they were placed in 3rd and 4th places, depending on the importance of criteria in the problem. Moreover, it is worth noting that when the importance of the accuracy factor was increased by 70%, the Serbia team in the second half and throws done after dribbling were the best-evaluated variant, while the second best was the Serbia team in the second half and throws done after passing. Thus, it could be seen that when the importance of accuracy increased in the evaluation process, the Serbia team's performance in the second half of the game was significantly better than that of the Poland team. The use of Technique for Order of Preference by Similarity to Ideal Solution (TOPSIS) and Relative Superiority Ratio (RSR) methods in similar studies [31] further illustrates how these analytical tools contribute to a deeper understanding of team dynamics and performance across different game situations. These methods allow us to quantify the immediate effects of tactical changes and project the potential impacts of strategic improvements over time. A similar method was used in the work of Hatem and Ikram (2023) [32] to create a new model for selecting starting lineups in football. This shows how the transfer of such tools can make it easier for the head coach to make decisions when selecting the best-matched players. Furthermore, it was evaluated how the assessed performance of the teams from the model would differ if the teams improved their performance by 5%, 10% and 20% regarding the initial values presented in the decision matrix. Thus, one value at a time for one analyzed variant was modified by a given percentage of improvement. We recorded the model's results, allowing us to see how improving shot efficiency and shot timing could impact a team's performance in two key areas of play. It could be seen that the most improvement noticed in the study was observed for the Serbia team in the second half and throws done after passing and 20% improvement ($A_7$), where the performance score increased by 0.153. The experiments showed that when teams could improve particular aspects of the game by a given percentage regarding the initial values, the Serbia team would still be assessed better, with an even more significant advantage than the Poland team. This is consistent with the work Kolbowicz et al. (2024) [33] that analyzes potential changes related to improving players' ratings in the rankings or changing performance, as shown Kizielewicz and Dobryakova (2020) [34]. Suggestions on how to improve the team, both in terms of selection and changes in tactics, can be of great importance in preparation for subsequent competitions of this type. These analytical findings illustrate potential pathways for teams to improve performance, as further discussed in the practical implications and conclusions.

**Practical application**

1. The results encourage coaches to develop dedicated and individualized training measures, emphasizing the ability to generate shooting actions for themselves and partners, enforcing adaptive processes to maximize the effectiveness of decisions made at key moments of the game.

2. The new evaluation template allows for additional analysis of the opponent. Thanks to this, coaches or players themselves can plan or create new, unusual tactics (new behaviours of the team, players) to reduce the effectiveness of the opponent's throws and thus increase their chance of winning.

3. Implementing a decision support model in sports allows for integrating multi-criteria analysis with the match preparation process by changing training requirements to identify key areas for improvement (future work).

## Limitations of the study

The sample size limits the study – only two national teams (Poland and Serbia) were analyzed in the group stage of the Olympic tournament. Further studies with more significant numbers of teams and analysis of subsequent tournaments may further confirm the obtained results and broaden the possibility of their generalization. Limitation is the use of basic statistical methods. Future studies should consider using more advanced statistical analysis techniques, such as multivariate analysis. It could show more precise relationships between variables, increase the analytical power of the results, and thus strengthen the scientific value of the obtained conclusions. Another limitation is the lack of comparison of the 3x3 basketball with the 5x5 version in various aspects. Future studies should consider the need for such an analysis

## Further research directions

1. The analysis should be extended to more teams and tournaments, using the current analysis model for continuity and a more complete picture of the impact of shooting strategies in 3x3 basketball.

2. Creation of a new model (parameters for assessing throws for 1 point) combining the analysis of tactics of all throws.

3. Studying the impact of players' fatigue and rotation strategies on the effectiveness of throws during the match and the entire tournament by taking into account locomotion data, e.g., Local Positioning Systems

4. Introduction to research on advanced technologies used to analyze the biomechanics of throws (body position in relation to the opponent's hoop, angle of throwing the ball, throwing technique) and video game strategies using algorithms defining specific behaviours on the pitch.

5. Development of a decision support model using a multi-criteria analysis tool for other team games - a universal approach.

## Conclusions

### 1. 2-point shots efficiency determines victory

a) Better 2-point shots efficiency significantly impacts success in 3x3 basketball; b) Teams with higher accuracy in this element have a better chance of winning; c) 2-point shots training should be a priority for coaches and players.

### 2. Key factors influencing shot efficiency

a) The position on the court from which the shot is taken directly affects its efficiency; b) Interaction with the defender: The defender's presence and actions can make it difficult to make an effective shot; c) The length of the action before the shot affects shooting decisions and precision.

### 3. Tactical flexibility is the key to success

a) Analysis of shooting strategies shows the differences between winning and losing teams; b) Adapting tactics during the game to the current situation is important; c) Tactical flexibility should be developed both in training and considered when selecting players.

## 4. New methods for optimizing game strategies

a) Decision modeling and multi-criteria analysis provide new tools for coaches and analysts; b) They help to evaluate and optimize game strategies; c) They make it easier to answer the question: "What should we change to win?"

In summary, despite the limited sample, this pilot analysis provides novel insights into 3x3 basketball strategy and offers a foundation for future research to build upon.

## Acknowledgments

The authors would like to thank all the people who provided support and advice at individual stages of the preparation of the manuscript.

## Author contributions

**Conceptualization:** Michał Nowak, Michał Skalik, Jakub Więckowski.

**Data curation:** Michał Nowak, Michał Skalik, Jakub Więckowski, Łukasz Oleksy.

**Formal analysis:** Michał Nowak, Michał Skalik, Jakub Więckowski.

**Funding acquisition:** Artur Stolarczyk, Łukasz Oleksy.

**Investigation:** Michał Nowak, Michał Skalik.

**Methodology:** Michał Nowak, Michał Skalik, Jakub Więckowski, Łukasz Oleksy.

**Project administration:** Michał Nowak.

**Resources:** Michał Skalik, Artur Stolarczyk.

**Software:** Michał Nowak, Michał Skalik, Jakub Więckowski, Radosław Ciejpa.

**Supervision:** Michał Nowak, Łukasz Oleksy.

**Visualization:** Michał Skalik, Jakub Więckowski, Radosław Ciejpa.

**Writing – original draft:** Michał Nowak, Jakub Więckowski.

**Writing – review & editing:** Michał Nowak, Radosław Ciejpa, Łukasz Oleksy.

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
