## [Decision Letter · Decision Letter 0]

16 Sep 2024

PONE-D-24-34221Winners' strategies: Comprehensive analysis and optimization of 2-point shots in 3x3 basketball using multi-criteria decision support analysis, on the example of two Olympic National Teams.PLOS ONE

Dear Dr. Nowak,

Thank you for submitting your manuscript to PLOS ONE. After careful consideration, we feel that it has merit but does not fully meet PLOS ONE’s publication criteria as it currently stands. Therefore, we invite you to submit a revised version of the manuscript that addresses the points raised during the review process.

**ACADEMIC EDITOR: **Dear authors, three independent reviewer had very positive comments on your study. However, they indicated some minor issues that should be addressed prior its final acceptance. 

We look forward to receiving your revised manuscript.

Kind regards,

Danica Janicijevic, Ph.D

Academic Editor

PLOS ONE

2. In the online submission form, you indicated that your data will be submitted to a repository upon acceptance. We strongly recommend all authors deposit their data before acceptance, as the process can be lengthy and hold up publication timelines. Please note that, though access restrictions are acceptable now, your entire minimal dataset will need to be made freely accessible if your manuscript is accepted for publication. This policy applies to all data except where public deposition would breach compliance with the protocol approved by your research ethics board. If you are unable to adhere to our open data policy, please kindly revise your statement to explain your reasoning and we will seek the editor's input on an exemption.

Additional Editor Comments (if provided):

Reviewers' comments:

Reviewer's Responses to Questions

**Comments to the Author**

1. Is the manuscript technically sound, and do the data support the conclusions?

Reviewer #1: Yes

Reviewer #2: Yes

Reviewer #3: Yes

2. Has the statistical analysis been performed appropriately and rigorously? 

Reviewer #1: Yes

Reviewer #2: I Don't Know

Reviewer #3: Yes

3. Have the authors made all data underlying the findings in their manuscript fully available?

Reviewer #1: Yes

Reviewer #2: Yes

Reviewer #3: Yes

4. Is the manuscript presented in an intelligible fashion and written in standard English?

Reviewer #1: Yes

Reviewer #2: No

Reviewer #3: Yes

5. Review Comments to the Author

Reviewer #1: The manuscript technically sound, and the data support the conclusion. Although it is well discussed, it is in much detailed detail, it could be summarized.

The authors made all data underlying the findings in their manuscript fully available.

The manuscript presented in an intelligible fashion and written in standard English.

Reviewer #2: Introduction is in overall well-written. Maybe there is a sufficient data about physical and motor status of 3x3 players comparing to the data related to the topic.

The name of the author in the reference no. 22 is Erculj instead of Ervculj.

Reviewer #3: The article presents an innovative approach to the strategic analysis of 2-point shots in 3x3 basketball, making a significant contribution to the literature in this field. The use of advanced analytical methods such as multi-criteria decision analysis (MCDA) enhances the value of the study in both academic and practical domains. When evaluated against PLOS ONE's publication criteria, the study has several strengths, including a title and abstract that reflect the content, and a methodology that is detailed and transparently presented. However, the methodological explanations need to include more information on participant selection and the ethical approval process. Additionally, the limited sample size and the lack of advanced statistical methods in data analysis introduce limitations in terms of the study's generalizability.

Presenting the results and discussion in a more concise and clearer manner would better align with PLOS ONE's policy of appealing to a broader readership. In terms of language and structure, the article generally uses scientific terminology effectively and has strong language, but the complexity of some sentences could be simplified to enhance clarity. A more comprehensive comparison with the literature in the discussion section would highlight the study’s original contribution and its alignment with other research.

Overall, the article is deemed suitable for publication in PLOS ONE, but it could be further strengthened with specific improvements. Such revisions would align with the literature emphasizing methodological clarity and transparency in data presentation within sports science, thereby enhancing the article's scientific value and the benefit it provides to the reader.

6. PLOS authors have the option to publish the peer review history of their article (what does this mean?). If published, this will include your full peer review and any attached files.

Reviewer #1: No

Reviewer #2: No

Reviewer #3: No

---

## [Author Response · Author response to Decision Letter 0]

12 Nov 2024

Responses to reviewers were placed in a separate file following the email instructions.

---

## [Editor Report · Decision Letter 1]

6 Feb 2025

PONE-D-24-34221R1Winners' strategies: Comprehensive analysis and optimization of 2-point shots in 3x3 basketball using multi-criteria decision support analysis, on the example of two Olympic National Teams.PLOS ONE

Dear Dr. Nowak,

Thank you for submitting your manuscript to PLOS ONE. After careful consideration, we feel that it has merit but does not fully meet PLOS ONE’s publication criteria as it currently stands. Therefore, we invite you to submit a revised version of the manuscript that addresses the points raised during the review process.

**ACADEMIC EDITOR: **

Dear Authors,

We are pleased to inform you that your article was reviewed by three independent reviewers. All reviewers expressed general satisfaction with your work, and only one reviewer suggested some minor changes, presented below. 

We look forward to receiving your revised manuscript.

Kind regards,

Danica Janicijevic, Ph.D

Academic Editor

PLOS ONE
---

## [Author Response · Author response to Decision Letter 1]

8 Mar 2025

All responses to reviewers were posted as instructed in the document

---

## [Editor Report · Decision Letter 2]

16 Mar 2025

Winners' strategies: Comprehensive analysis and optimization of 2-point shots in 3x3 basketball using multi-criteria decision support analysis, on the example of two Olympic National Teams.

PONE-D-24-34221R2

Dear Dr. Michal Jakub Nowak,

We’re pleased to inform you that your manuscript has been judged scientifically suitable for publication and will be formally accepted for publication once it meets all outstanding technical requirements.

Kind regards,

Danica Janicijevic, Ph.D

Academic Editor

PLOS ONE
---

## [Editor Report · Acceptance letter]

PONE-D-24-34221R2

PLOS ONE

Dear Dr. Nowak,

I'm pleased to inform you that your manuscript has been deemed suitable for publication in PLOS ONE. Congratulations! Your manuscript is now being handed over to our production team.

Kind regards,

on behalf of

Dr. Danica Janicijevic

Academic Editor

PLOS ONE